# Prevalence of SARS-CoV-2 Antibodies in Multiple Sclerosis: The Hidden Part of the Iceberg

**DOI:** 10.3390/jcm9124066

**Published:** 2020-12-16

**Authors:** Nicola Capasso, Raffaele Palladino, Emma Montella, Francesca Pennino, Roberta Lanzillo, Antonio Carotenuto, Maria Petracca, Rosa Iodice, Aniello Iovino, Francesco Aruta, Viviana Pastore, Antonio Riccardo Buonomo, Emanuela Zappulo, Ivan Gentile, Maria Triassi, Vincenzo Brescia Morra, Marcello Moccia

**Affiliations:** 1Multiple Sclerosis Clinical Care and Research Centre, Department of Neuroscience, Reproductive Sciences and Odontostomatology, University of Naples “Federico II”, 80138 Naples, Italy; nicolacapasso91@gmail.com (N.C.); robertalanzillo@libero.it (R.L.); carotenuto.antonio87@gmail.com (A.C.); maria@petraccas.it (M.P.); rosa.iodice@unina.it (R.I.); anielloiovino@msn.com (A.I.); fra.aruta92@gmail.com (F.A.); vincenzo.bresciamorra2@unina.it (V.B.M.); 2Department of Public Health, University of Naples “Federico II”, 80138 Naples, Italy; raffaele.palladino@unina.it (R.P.); francesca.pennino@unina.it (F.P.); vivianapastore10@gmail.com (V.P.); triassi@unina.it (M.T.); 3Department of Primary Care and Public Health, Imperial College London, London W68RP, UK; 4Department of Hygiene, Preventive and Industrial Medicine, University Hospital “Federico II”, 80138 Naples, Italy; emma.montella@unina.it; 5Section of Infectious Diseases, Department of Clinical Medicine and Surgery, University of Naples “Federico II”, 80138 Naples, Italy; antonioriccardobuonomo@gmail.com (A.R.B.); e.zappulo@gmail.com (E.Z.); ivan.gentile@unina.it (I.G.); 6UNESCO Chair on Health Education and Sustainable Development, University of Naples “Federico II”, 80138 Naples, Italy

**Keywords:** multiple sclerosis, COVID-19, infection, antibody, seroprevalence

## Abstract

Background. We compared the prevalence of SARS-CoV-2 IgG/IgM in multiple sclerosis (MS), low-risk, and high-risk populations and explored possible clinical correlates. Methods. In this cross-sectional study, we recruited MS patients, low-risk (university staff from non-clinical departments), and high-risk individuals (healthcare staff from COVID-19 wards) from 11 May to 15 June 2020. We used lateral flow immunoassay to detect SARS-CoV-2 IgG and IgM. We used t-test, Fisher’s exact test, chi square test, or McNemar’s test, as appropriate, to evaluate between-group differences. Results. We recruited 310 MS patients (42.3 ± 12.4 years; females 67.1%), 862 low-risk individuals (42.9 ± 13.3 years; females 47.8%), and 235 high-risk individuals (39.4 ± 10.9 years; females 54.5%). The prevalence of SARS-CoV-2 IgG/IgM in MS patients (*n* = 9, 2.9%) was significantly lower than in the high-risk population (*n* = 25, 10.6%) (*p* < 0.001), and similar to the low-risk population (*n* = 11, 1.3%) (*p* = 0.057); these results were also confirmed after random matching by age and sex (1:1:1). No significant differences were found in demographic, clinical, treatment, and laboratory features. Among MS patients positive to SARS-CoV-2 IgG/IgM (*n* = 9), only two patients retrospectively reported mild and short-lasting COVID-19 symptoms. Conclusions. MS patients have similar risk of SARS-CoV-2 infection to the general population, and can be asymptomatic from COVID-19, also if using treatments with systemic immunosuppression.

## 1. Introduction

On 11 March 2020, the World Health Organization declared coronavirus disease-19 (COVID-19) from severe acute respiratory syndrome coronavirus (SARS-CoV-2) to be a pandemic. In the following months, COVID-19 has caused severe morbidity and mortality worldwide, especially in elderly populations, in males, and in individuals with concomitant diseases (e.g., heart disease, diabetes) [1].

People with multiple sclerosis (MS) might be especially at risk from COVID-19 morbidity and mortality because they are known to suffer and to die more frequently from respiratory and infectious diseases [2]. Additionally, people with MS have higher prevalence of a number of comorbidities (e.g., cardiovascular) when compared with the general population [3,4]. Not least, the use disease-modifying treatments (DMTs), causing different degrees of systemic immunosuppression, might further affect the possibility of preventing and responding to the infection [5]. In line with this, healthcare providers and policy makers have immediately advised people with MS to self-isolate, and the use of immunomodulatory and immunosuppressive DMTs has been postponed [6,7,8].

Serological tests to detect IgM and IgG immunity, which increase from the second week after COVID-19 onset of symptoms [9], have been used for epidemiological purposes [10] because they can identify all infected individuals, including those with no or mild symptoms [10,11]. In particular, asymptomatic carriers can transmit the virus in the absence of obvious symptoms and could be responsible for keeping SARS-CoV-2 circulating [12]. As such, studying the prevalence of SARS-CoV-2 IgG/IgM in MS can shed light on the risk of COVID-19 infection in relation to MS and/or to the use of some DMTs, on the amount of patients who are still susceptible to infection, and on the possibility of asymptomatic carriers in MS, and can also be used to plan clinical activities accordingly [5,8,13,14,15]. Thus, in the present study, we aim to: (1) evaluate the prevalence of SARS-CoV-2 IgG/IgM antibodies in asymptomatic MS patients, compared with populations at low-risk and high-risk of COVID-19 infections; and (2) explore possible correlates with demographics, clinical features, treatments, comorbidities, and laboratory findings.

## 2. Methods

### 2.1. Study Design

This is a cross-sectional study including all MS patients attending the MS Clinical Care and Research Centre at Federico II University Hospital of Naples (Campania Region, Italy) from 11 May to 15 June 2020. For comparison purposes, we also included a sample of low-risk individuals (university staff from non-clinical departments) and a sample of high-risk individuals (healthcare staff from COVID-19 wards), who underwent the same serological test during the same period.

The study was approved by the Federico II Ethics Committee (355/19 and subsequent amendments). All patients signed informed consent authorizing the use of anonymized data collected routinely as part of clinical practice, in line with data protection regulations (GDPR EU2016/679). The study was performed in accordance with good clinical practice and the Declaration of Helsinki.

### 2.2. Study Population

During the study period (11 May to 15 June 2020), the COVID-19 Task Force from our University made SARS-CoV-2 IgG/IgM antibody testing compulsory for all patients attending the hospital, for healthcare staff from COVID-19 wards, and for university staff required to come back to office work following lockdown. This policy was driven by the possibility of using serological data to safely deploy healthcare workers, to reduce exposure to the virus in susceptible individuals, and to assess the effect of lockdown at population level [11]. Because this policy was time-limited, we selected the period of time when all individuals were tested; all individuals then consented to study inclusion.

For MS patients, inclusion criteria were: (1) diagnosis of MS [16]; (2) consent to participate; (3) scheduled consultation at the MS Centre from 11 May to 15 June 2020; (4) residence in the Campania Region of Italy.

For low-risk individuals, inclusion criteria were: (1) university staff from non-clinical departments who had been self-isolating at home during lockdown; (2) consent to participate; (3) no history of MS or other central nervous system diseases; (4) no history of chronic diseases and treatments; (5) residence in the Campania Region of Italy.

For high-risk individuals, inclusion criteria were: (1) healthcare staff from COVID-19 wards; (2) consent to participate; (3) no history of MS or other central nervous system diseases; (4) no history of chronic diseases and treatments; (5) residence in the Campania Region of Italy.

For MS patients, low-risk individuals, and high-risk individuals, exclusion criteria were: (1) age <18 years; (2) incomplete records; (3) previous COVID-19 diagnosis; (4) COVID-19 symptoms (e.g., cough, fever, anosmia, difficulty breathing), either active or in the past 14 days. In particular, as per University policy, all individuals with symptoms of COVID-19 (e.g., cough, fever, anosmia, difficulty breathing), either active or in the past 14 days, were denied access to the University and to the testing.

### 2.3. Antibody Detection

We used lateral flow immunoassay (LFIA) to detect SARS-CoV-2 IgG and IgM antibodies (Shanghai Kehua Bio-engineering Co., Ltd., Shanghai, China), in accordance with manufacture instructions [17]. LFIA is a rapid method based on immunochromatography that uses colloidal gold conjugated COVID-19 antigens. It comprises a plastic pad where a nitrocellulose membrane is fitted. Three separate lines are created by immobilizing goat anti-human IgM, IgG and goat anti-rabbit-IgG to assess the presence of IgM, IgG, and control (C) lines, respectively. The entire conjugate pad is sprayed with a mixture of AuNP-COVID-19 recombinant antigen-conjugate (colloidal-gold pre-treated with SARS-CoV-2 recombinant protein) and AuNP-rabbit-IgG. A blood sample is applied to the sample pad and, with the aid of a buffer, migrates towards the immobilized lines of antibodies spread with the AuNP-recombinant antigen. When a reaction occurs, a visible line is formed, suggesting the existence of IgM and/or IgG, separately; color in the control line should be formed for a test to be valid [17,18]. The test provides a qualitative result, which was visually examined by a nurse and a physician together (unblinded to group status) 15 min after blood sample application. Though far from perfect (66% pooled sensitivity, 96.6% pooled specificity), LFIA performs the best at our expectedly low prevalence rates (<5%), and is especially indicated for screening purposes [19]. For statistical purposes, we combined IgM and IgG to define SARS-CoV-2 IgG/IgM positive individuals.

### 2.4. Demographics, Clinical Features, Treatments, and Laboratory Findings

For MS patients, we used clinical records to retrieve age, sex, and expanded disability status scale (EDSS), current disease modifying treatments (DMTs), comorbidities, and most recent laboratory findings (within one month from SARS-CoV-2 IgG/IgM testing) for white blood cell count, total lymphocyte count, and lactic dehydrogenase (these were specifically selected because they have been reported to be commonly changed in COVID-19) [20]. For statistical purposes, we classified DMTs based on the suggested risk of systemic immunosuppression [6,7].

During the same visit, for MS patients who tested positive to SARS-CoV-2 IgG/IgM, we used a standard clinical questionnaire to investigate the occurrence of COVID-19 symptoms (e.g., cough, fever, anosmia, difficulty breathing), and any possible at-risk behavior (e.g., contact with defined COVID-19 cases, travel to high COVID-19 prevalence areas) in the previous three months (corresponding to the beginning of the first wave of COVID-19 infections in the area).

For low-risk and high-risk individuals, we recorded age, sex, and SARS-CoV-2 IgG/IgM status.

### 2.5. Sample Size Calculation

Based on published data, we estimated that approximately 5500 individuals with MS live in the Campania Region of Italy (among which 30–40% were followed up at our center) [21]. Assuming COVID-19 prevalence ranges from 1% to 10% [22], a sample of 300 MS patients would be enough to estimate COVID-19 prevalence in the MS population with 5% precision and 95% confidence intervals.

### 2.6. Statistics

To evaluate differences in prevalence of SARS-CoV-2 IgG/IgM between MS, low-risk, and high-risk populations (aim 1), we first calculated raw prevalence. Differences in prevalence rates were further assessed after performing a random matching by age and sex (1:1:1), considering that the three populations (MS, low-risk, and high-risk) had different age and sex distribution. Differences in prevalence were evaluated using chi square test, Fisher’s exact test, or McNemar’s test, as appropriate.

To evaluate differences in demographics, clinical features and laboratory findings between MS patients with or without SARS-CoV-2 IgG/IgM (aim 2), we used *t*-test, chi square test, or Fisher’s exact test, as appropriate.

Statistical analyses were performed using Stata 15.0. Following Bonferroni correction for multiple comparisons, results were considered statistically significant for *p* < 0.005.

### 2.7. Data Availability

Data supporting the findings of this study are available if requested to the authors.

## 3. Results

We recruited 310 MS patients, 862 low-risk individuals, and 235 high-risk individuals. Demographics and SARS-CoV-2 IgG/IgM status are reported in Table 1.

The raw prevalence of SARS-CoV-2 IgG/IgM in MS (*n* = 9, 2.9%) was significantly lower than in the high-risk population (*n* = 25, 10.6%) (*p* < 0.001) and similar to the low-risk population (*n* = 11, 1.3%) (*p* = 0.057). Similarly, after random matching by age and sex, MS, low-risk, and high-risk populations (148 individuals in each group), the prevalence of SARS-CoV-2 IgG/IgM in MS (*n* = 6, 4.0%) was significantly lower than in the high-risk population (*n* = 20, 13.5%) (*p* = 0.001) and similar to the low-risk population (*n* = 1, 0.7%) (*p* = 0.130).

Demographics, clinical features, treatments, and laboratory findings of MS patients are reported in Table 2. MS patients positive to SARS-CoV-2 IgG/IgM were similar to patients negative to SARS-CoV-2 IgG/IgM in age (*p* = 0.830), sex (*p* = 0.988), EDSS (*p* = 0.642), DMTs (*p* = 0.486), comorbidities (*p* = 0.605), white blood cell count (*p* = 0.301), total lymphocyte count (*p* = 0.129), and lactic dehydrogenase (*p* = 0.452).

Demographics, clinical features, treatments, and laboratory findings of MS patients positive to SARS-CoV-2 IgG/IgM are detailed in Table 3. Only two patients retrospectively reported on possible COVID-19 symptoms, but, at that time, did not undergo SARS-CoV-2 nasopharyngeal-oropharyngeal swab; in particular, one patient reported a cough, which only lasted a few days, and another patient, who had travelled to a high COVID-19 prevalence area (Switzerland through the north of Italy), reported a fever (below 38.5 °C) and anosmia that, again, only lasted a few days. Patients presented with a wide range of clinical disability (EDSS was from 1.0 to 6.5). Eight patients were currently under the clinical effect of DMTs (alemtuzumab, cladribine, natalizumab, and teriflunomide). Looking at laboratory tests, we found values below normal limits for white blood cell count in one patient and for lymphocytes in three patients (one patient had grade 1 lymphopenia (0.8–1 × 10^3^/μL) and two patients had grade 2 lymphopenia (0.5–0.7 × 10^3^/μL)); lactic dehydrogenase was within normal limits in all patients. Following positive SARS-CoV-2 IgG/IgM testing, all patients underwent SARS-CoV-2 nasopharyngeal-oropharyngeal swab, which resulted negative for SARS-CoV-2 RNA, suggesting previous rather than active infection.

## 4. Discussion

In our population, MS patients, in the absence of overt COVID-19 symptoms, did not present with a significantly higher prevalence of SARS-CoV-2 antibodies when compared with low-risk individuals (2.9% vs. 1.3%). As such, MS patients can develop SARS-CoV-2 immunity, following mild or no COVID-19 symptoms also if using DMTs with high risk of systemic immunosuppression. No significant correlates (e.g., demographics, clinical features, treatments, and laboratory findings) were found for the prevalence of SARS-CoV-2 IgG/IgM in MS.

We specifically set a time frame (11 May to 15 June 2020) and geographical area (Campania Region of Italy) for the three populations in order to study a community undergoing similar infection distribution and lockdown policies; for instance, the conduction of the study within three months from the beginning of the local epidemic reduced the risk of underestimating the prevalence of SARS-CoV-2 IgG/IgM, due to the possibility of antibodies vanishing after this time frame [9]. During the first wave, the Campania Region of Italy was considered a low-prevalence area for COVID-19 [22]. A recent report of the Italian institute of Statistics showed <1% seroprevalence in the Campania region at the time our study was conducted. Accordingly, we found 1.3% seroprevalence in healthy individuals who self-isolated during the lockdown using LFIA, which is at the higher risk of false positive results when compared with CLIA/ELISA, used by the Italian Institute of Statistics [23]. Overall, our estimates are not far from other low-prevalence areas (e.g., 1.79% in Boise, ID, USA) [22,24], and lower than high-prevalence areas (e.g., 4.65% in Los Angeles County, CA, USA) [22,25]. As expected, the prevalence of SARS-CoV-2 IgG/IgM in low-risk individuals (1.3%) and MS (2.9%) was not as high as in high-risk individuals (10.6%). In particular, we selected healthcare staff from COVID-19 wards as a high-risk reference, following previous studies showing high SARS-CoV-2 IgG/IgM in this population [26,27].

The numerically higher prevalence of SARS-CoV-2 IgG/IgM in MS, when compared with low-risk populations, is not surprising. Indeed, MS patients, though possibly worried about risks coming from COVID-19 to individuals with comorbidities, as from national and international recommendations [1,20], have inevitably had hospital access for disease and treatment management during the lockdown, with subsequent risks of COVID-19 infection. Noteworthy, people with MS have been initially classified more at risk of COVID-19 morbidity and mortality [1], and the exposure to DMTs with systemic immunosuppression is thought to further increase the risk of infections [28], though this was not disclosed in our population, possibly due to sample size constraints. Additionally, we did not find any differences in demographics, clinical features, and laboratory findings suggestive of COVID-19 infection in MS patients positive and negative to SARS-CoV-2 IgG/IgM [20].

COVID-19 pandemic has represented a new challenge for neurologists treating patients with MS, as a consequence of possible risks coming from this infection to these patients [5,6,14]. We showed that seven out of nine MS patients using immunomodulatory and immunosuppressive DMTs presented with no symptoms from COVID-19, and two only had mild symptoms. This finding is in line with results on the general population, where up to four fifths of individuals are expected to stay asymptomatic [12], and with previous reports suggesting that MS patients using DMTs have standard COVID-19 morbidity and mortality [29,30,31]. Unfortunately, the presence of past COVID-19 symptoms was not investigated routinely in our control populations, though they did not have previous COVID-19 diagnosis nor had active/recent respiratory symptoms (as from inclusion criteria). Interestingly, in a UK community-based study including 3907 MS patients and using a questionnaire for self-reported diagnosis, Evangelou and colleagues showed that MS patients (and the use of DMTs) were not at increased risks of COVID-19 [32]. Similarly, French and Dutch studies did not find any association between DMT and COVID-19 severity [33,34]. In another previous study, the Chinese Medical Network for Neuroinflammation conducted a survey on 1804 MS patients and found no cases with formal COVID-19 diagnosis, irrespective of DMTs [35]. However, considering that during the first wave of the pandemic healthcare systems have struggled to guarantee medical care to moderate-severe cases [22], it is possible that MS patients with mild or no COVID-19 symptoms have gone undiagnosed, thus suggesting that future prevalence studies should combine different diagnostic modalities (e.g., clinical history, formal diagnosis of COVID-19, seroprevalence).

Diagnosis of active COVID-19 infection is currently based on nasopharyngeal-oropharyngeal swab and real-time polymerase chain reaction (RT-PCR), which is however unable to detect past infections in prevalence studies [11] and has also a potentially high false negative rate [36]. On the contrary, antibody testing for COVID-19 can support a diagnosis of both active and past infections [13]. A large number of tests have been developed for COVID-19 antibody detection, such as LFIA (used in the present study), chemiluminescent immunoassay (CLIA), enzyme-linked immunosorbent assay (ELISA), and Fluorescence Immunoassays (FIA), all of them targeting IgG and/or IgM antibodies against S and/or N viral proteins of human serum/blood samples [17,18,36,37]. A recent meta-analysis has showed that ELISA and LFIA have the highest specificity, reaching levels >99%, whilst ELISA and CLIA performed better in terms of sensitivity (90–96%), followed by LFIA and FIA with sensitivities ranging from 80% to 89% [10,17]. As such, our test (LFIA) is potentially at risk of false positive results; however, we included MS cases and two control populations, among which the risk of false positive results should have been equally distributed [19]. Overall, we preferred LFIA because it can be quickly used on-site and is particularly attractive for large seroprevalence studies as a consequence of high specificity (e.g., reduced risk of missing positive cases). At the individual level, however, mixed strategies should be adopted (e.g., re-testing a positive case using a different serological test and/or nasopharyngeal-oropharyngeal swab) [10], and, in keeping with this, in MS patients positive to SARS-CoV-2 IgG/IgM, we excluded the presence of active COVID-19 infection with nasopharyngeal-oropharyngeal swab.

The present study raises a number of policy implications for clinical practice. First, there are risks coming from asymptomatic MS patients that could be responsible for COVID-19 spreading to healthcare staff and other patients at MS Centers [12]. Not least, whilst the use of DMTs is generally contraindicated in patients with symptoms of active infection, the use of DMTs causing acute systemic immunosuppression in COVID-19 asymptomatic patients could possibly increase morbidity and mortality, as for the reactivation of chronic asymptomatic infections (e.g., hepatitis, tuberculosis, herpes viruses) [38]. Additionally, the prevalence of SARS-CoV-2 IgG/IgM antibodies in MS is low, and, thus, this population is especially vulnerable to future COVID-19 infections in the case of additional epidemic waves and/or local outbreaks. Safety protocols for accessing the MS centre and for DMT dosing/redosing should then account for additional risks coming from COVID-19. A combination of clinical history (e.g., at-risk contacts, presence of active symptoms), testing (e.g., serology, swab), and non-pharmacological measures (e.g., face masks, self-isolation before/after some DMTs) should be carefully considered within national and international recommendations for MS healthcare organization and clinical practice [5]. Finally, we have also shown the possibility of developing COVID-19 antibodies in MS patients, also if lymphocyte levels were below normal values following DMTs with systemic immunosuppression, suggesting this population would be suitable for vaccination, when available.

Among the study limitations, there is the low sensitivity of SARS-CoV-2 IgG/IgM LFIA testing, with risk of false-positive results in low-incidence settings [10]. False negatives could also have occurred in previously-infected patients who failed to produce antibodies specific to COVID-19 antigens, in patients where antibodies quickly waned, or in patients who have not mounted a specific antibody response yet. However, we performed the same test to both MS patients and controls from the same geographical area and within the same time period and, thus, the risk of false negatives applies equally to different subgroups. Of note, the subpopulation treated with anti-CD20 medications could not have developed antibodies and, therefore, tested negative in our study; this subpopulation could also be more at risk of worse SARS-CoV-2 outcomes and, thus, could have been excluded from the present study that specifically focused on patients with no symptoms/undetected infection. LFIA has already been used for SARS-CoV-2 IgG/IgM prevalence studies worldwide [25] and, considering that IgM and IgG tests have different sensitivity in relation to the phases of the infection (e.g., IgM antibodies increase earlier than IgG) [9], we have combined IgG and IgM testing to improve sensitivity [10]. Our study was cross-sectional and was run within clinical practice, in the absence of actual follow-up to study seroconversion prospectively. We were not able to perform exact matching for the whole populations due to sample size constraints; still, our results were consistent both on raw prevalence estimates and after age- and sex-matching. Our single-center recruitment holds a potential selection bias and, for instance, we cannot exclude the possibility that some older and more disabled patients might have missed the appointment while sheltering at home. Additionally, due to local regulations, hospital access was limited to patients with more complex healthcare needs (e.g., infusion therapy). However, our population is numerically representative of MS patients living in the Campania Region of Italy, and also has a similar age (42 vs. 44 years) [21]. Finally, we were not fully aware of the working conditions and compliance with lockdown/self-isolation policies of our population.

In conclusion, MS patients are not at high risk of SARS-CoV-2 infection, though the prevalence of SARS-CoV-2 IgG/IgM is numerically higher than age- and sex-matched low risk individuals. As in the general population, most MS patients positive to SARS-CoV-2 IgG/IgM did not report on any COVID-19 symptom, also if using treatments with high risk of systemic immunosuppression. In clinical practice, healthcare staff should account for the additional risks coming from COVID-19, while delivering regular care.

## Figures and Tables

**Table 1 jcm-09-04066-t001:** Demographics and SARS-CoV-2 IgG/IgM status in multiple sclerosis (MS) patients, low-risk population, and high-risk population.

	MS Patients (*n* = 310)	Low-Risk Population (*n* = 862)	High-Risk Population (*n* = 235)
Age, years	42.3 ± 12.4	42.9 ± 13.3	39.4 ± 10.9
Sex, females (%)	208 (67.1%)	412 (47.8%)	128 (54.5%)
SARS-CoV-2 status, number (prevalence %)			
IgG or IgM positive	9 (2.9%)	11 (1.3%)	25 (10.6%)
Females	6 (2.9%)	7 (1.7%)	14 (10.9%)
Males	3 (2.9%)	4 (0.9%)	11 (10.3%)
IgG positive	9 (2.9%)	5 (0.6%)	9 (3.8%)
IgM positive	0 (0%)	6 (0.7%)	8 (3.4%)
IgM and IgG positive	0 (0%)	0 (0%)	8 (3.4%)

**Table 2 jcm-09-04066-t002:** Demographics, clinical features, treatments, and laboratory findings of MS patients.

	MS Negative to SARS-CoV-2 IgG/IgM (*n* = 301)	MS Positive to SARS-CoV-2 IgG/IgM (*n* = 9)
Age, years	42.2 ± 12.4	41.4 ± 12.8
Females, number (%)	202 (67.1%)	6 (66.6%)
Expanded disability status scale (EDSS), median (range)	3.5 (0–8.0)	3.0 (1.0–6.5)
Disease-modifying treatments (DMT), number (%)		
No/Low risk of systemic immunosuppression	187 (62.1%)	5 (55.6%)
Dimethyl Fumarate	8	0
Interferon	3	0
Glatiramer	1	0
Natalizumab	166	3
Teriflunomide	3	1
No DMT	7	1
Moderate/high risk of systemic immunosuppression	114 (37.9%)	4 (44.4%)
Alemtuzumab	16	3
Cladribine	5	1
Fingolimod	12	0
Ocrelizumab	80	0
Rituximab	1	0
Comorbidities, number (%)	32 (10.6%)	0 (0%)
Diabetes	1	0
High blood pressure	19	0
High cholesterol	9	0
Thyroid disease	7	0
White blood cell count, ×10^3^/μL	7.43 ± 2.32	6.62 ± 2.07
Total lymphocyte count, ×10^3^/μL	2.48 ± 1.35	1.79 ± 1.01
Lactic dehydrogenase, U/L	215.83 ± 49.94	223.44 ± 52.30

**Table 3 jcm-09-04066-t003:** Demographics, clinical features, treatments, and laboratory findings of MS patients positive to SARS-CoV-2 IgG/IgM in May–June 2020.

	1	2	3	4	5	6	7	8	9
Age, years	39	54	42	27	35	65	38	57	29
Sex	Male	Female	Female	Female	Female	Female	Female	Male	Male
COVID-19 symptoms	None	Cough	None	None	None	None	None	None	Fever, anosmia
COVID-19 at-risk behaviour	None	None	None	None	None	None	None	None	Travel
EDSS	2.5	3.0	6.5	1.0	4.5	2.5	4.5	4.0	1.5
DMT	Alemtuzumab	Natalizumab	Alemtuzumab	Natalizumab	Natalizumab	Teriflunomide	Alemtuzumab	None	Cladribine
Last DMT administration	January 2019	February 2020	June 2018	March 2020	February 2020	Ongoing	January 2018		July 2019
Comorbidities	None	None	None	None	None	None	None	None	None
White blood cell count *, ×10^3^/μL	10.27	7.56	7.45	6.93	5.6	7.17	7.29	3.25	4.12
Total lymphocyte count **, ×10^3^/μL	1.81	2.85	0.90	3.18	3.06	1.97	1.61	0.69	0.71
Lactic dehydrogenase ***, U/L	274	225	215	178	190	332	172	239	186

* Normal range 4–10 × 10^3^/μL; ** Normal range 1–4.8 × 10^3^/μL; *** Normal range 140–280 U/L.

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
