# Peer review of "Prevalence of SARS-CoV-2 Antibodies in Multiple Sclerosis: The Hidden Part of the Iceberg"

_jcm, 2020, doi:10.3390/jcm9124066_

Round 1
Reviewer 1 Report
The authors took advantage of an excellent opportunity to study the question of "risk" to their MS population compared to LOW and HIGH risk groups. They were adequately powered to address this question, but not to look at whether certain MS DMT conferred greater risk or not and that is a limitation they should declare.
Most (80%) of the patients were on either natalizumab (the majority) or ocrelizumab compared with other DMT, so this was a skewing of the population not typical of most MS clinics. Authors should state that too as a limitation.
For the patients on ocrelizumab or rituximab, could these agents not produce false negative results based on their effect on B/plasma cells making the antibodies? If so, that too should be listed as a limitation. It is odd that other reports indicate a higher risk for these patients to contract COVID, but not a single one did so in this study. Again, the numbers (81) were probably too small.
Author Response
- The authors took advantage of an excellent opportunity to study the question of "risk" to their MS population compared to LOW and HIGH risk groups. They were adequately powered to address this question, but not to look at whether certain MS DMT conferred greater risk or not and that is a limitation they should declare.
We thank the Reviewer for this comment.
- Most (80%) of the patients were on either natalizumab (the majority) or ocrelizumab compared with other DMT, so this was a skewing of the population not typical of most MS clinics. Authors should state that too as a limitation.
As suggested, we have now added this sentence in the limitation section of the Discussion:
“Also, due to local regulations, hospital access was limited to patients with more complex healthcare needs (e.g., infusion therapy).”
For the patients on ocrelizumab or rituximab, could these agents not produce false negative results based on their effect on B/plasma cells making the antibodies? If so, that too should be listed as a limitation. It is odd that other reports indicate a higher risk for these patients to contract COVID, but not a single one did so in this study. Again, the numbers (81) were probably too small.
As suggested, we have now added the following sentences in the Discussion:
“Of note, the subpopulation treated with anti-CD20 medications could have not developed antibodies and, so, tested negative in our study; this subpopulation could be also more at risk of worse SARS-CoV-2 outcomes and, thus, could have been excluded from the present study which specifically focused on patients with no symptoms/undetected infection.”
Reviewer 2 Report
The authors have described SARS-CoV-2 IgG/IgM testing using a lateral flow immunoassay in three cohorts. This approach has several limitations which the authors acknowledge. The data is of interest although the numbers of positives were low and this may reflect a lack of sensitivity of the technique used. For the benefit of readers (subject to the availability of such data), it may be useful to provide estimates of disease activity in populations of this region based on clinical criteria and nucleic acid testing.
The authors' analyses and discussion of the data are comprehensive and the presentation is of an adequate standard. There is not much else for me to say.
Author Response
- The authors have described SARS-CoV-2 IgG/IgM testing using a lateral flow immunoassay in three cohorts. This approach has several limitations which the authors acknowledge. The data is of interest although the numbers of positives were low and this may reflect a lack of sensitivity of the technique used. For the benefit of readers (subject to the availability of such data), it may be useful to provide estimates of disease activity in populations of this region based on clinical criteria and nucleic acid testing. The authors' analyses and discussion of the data are comprehensive and the presentation is of an adequate standard. There is not much else for me to say.
We thank the Reviewer for his/her comments. We have now referenced to the recent report of the Italian Institute of Statistics which shows <1% seroprevalence in the Campania region at the time our study was conducted. This is not much different from our seroprevalence estimates, which could have also affected by the higher risk of false positive results of LFIA, when compared with CLIA/ELISA used by the Italian Institute of Statistics.